# A Questionnaire Survey on Long-Term Outcomes in Cats Breed-Screened for Feline Cardiomyopathy

**DOI:** 10.3390/ani12202782

**Published:** 2022-10-15

**Authors:** Anna Follby, Anna Pettersson, Ingrid Ljungvall, Åsa Ohlsson, Jens Häggström

**Affiliations:** 1AniCura Läckeby Djursjukhus, SE-395 98 Läckeby, Sweden; 2AniCura Djursjukhuset i Jönköping, SE-554 75 Jönköping, Sweden; 3Department of Clinical Science, Faculty of Veterinary Medicine and Animal Science, Swedish University of Agricultural Sciences, SE-750 07 Uppsala, Sweden; 4Department of Animal Breeding and Genetics, Faculty of Veterinary Medicine and Animal Science, Swedish University of Agricultural Sciences, SE-750 07 Uppsala, Sweden

**Keywords:** heart disease, cardiomyopathy, morbidity, mortality, survival

## Abstract

**Simple Summary:**

Feline cardiomyopathy (FCM) is a serious, potentially fatal disease in cats. There is an international screening program that aims to identify pedigree cats affected with FCM, as the disease is believed to be inherited in some cat families. Using a self-reporting questionnaire, this study explored the long-term outcomes of cats breed-screened for FCM. We found that approximately 9.3% of the cats developed FCM at some time-point of which approximately 50% were diagnosed within the screening program and 50% of these cats at the first breed-screen occasion. For cats that did develop FCM, there was a significantly higher risk for a cardiac related death and also a shorter time to all-cause mortality, compared to cats that did not develop FCM. Frequency and types of non-cardiac disease were similar in all screen classification groups. The large proportion of cats that did develop FCM later in life, despite normal previous screen results, underscores the value of repeated breed-screenings later in life to identify cats that develop FCM.

**Abstract:**

Feline cardiomyopathy (FCM) is an important contributor to feline morbidity and mortality. This explorative follow-up questionnaire study was aimed at investigating the long-term outcome in cats breed-screened for FCM (BS-FCM) in three Nordic countries. Records of cats with ≥1 BS-FCM between 2004–2015 were included. Of the 1113 included cats, 104/1113 (9.3%) had developed FCM at some time-point. Fifty-nine of the 104 (56.7%) FCM cats were diagnosed within the screening program (Screen^FCM^), and 33/59 (55.9%) of these were diagnosed at the first BS-FCM. Screen^FCM^ cats or with an owner-reported FCM diagnosis at a later time-point had a higher risk of cardiac-related death compared to cats that never developed FCM. A shorter lifespan was found in Screen^FCM^ cats compared to those with normal screen results (*p* < 0.001). Times to all-cause mortality were shorter (*p* < 0.001) in cats that developed FCM at any time-point compared to those that did not. Non-cardiac morbidities were similar in all screen classification groups. The large proportion of cats that developed FCM at a later time-point underscores the need for repeated screenings later in life. Cats that developed FCM at any time-point had a shorter lifespan, with a similar proportion and in line with the nature of non-cardiac morbidities, compared to those without FCM.

## 1. Introduction

Feline cardiomyopathy (FCM) is one of the most important contributors to feline morbidity and mortality [1,2,3,4,5,6,7,8,9]. Cardiomyopathy is defined as a disease of the myocardium associated with cardiac dysfunction and abnormal heart muscle structure, with no other concurrent cardiovascular disease. Cardiomyopathies include phenotypic categories of both known and unknown causes, and are traditionally subdivided, depending on structural and functional features, into hypertrophic—(HCM), restrictive—(RCM), dilated and arrhythmogenic right ventricular cardiomyopathy, and non-specific phenotypes [10,11,12]. It is currently not known whether these phenotypes are related to each other, or whether they represent distinct entities. Furthermore, the phenotype might change from one form to another over time, due to disease progression, comorbidities, or unknown factors [10]. Congestive heart failure (CHF), arterial thromboembolism (ATE) and sudden cardiac death are well-recognized complications to FCM, although many cats with preclinical FCM, particularly those with the preclinical hypertrophic phenotype, can live for a long time [13,14,15,16,17,18,19,20,21,22,23,24,25,26,27,28]. A recent study concluded that cats with cardiovascular morbidity had a shorter survival than those that did not [14].

The most common FCM phenotype is HCM [5,6,7,8,16,17,21,22,23,24,25,26,27,28,29,30,31] which is believed to be inherited as an autosomal dominant trait in some cat families, and screening for breeding purposes is, accordingly, becoming more commonly practiced [29,32,33,34]. By removing affected cats from breeding, the incidence of FCM might be reduced. An international screening and breeding program for FCM was initiated in 2004 and is today administered through the PawPeds program (www.pawpeds.com accessed on 10 January 2022). This program aims to identify pedigree cats affected with FCM and to recommend breeders to remove them from breeding. Owners are encouraged to screen their cat’s hearts at repeated times [32], but the long-term outcomes of breed-screened cats are hitherto unknown. Thus, the aim of the present study was to explore the long-term outcomes for cats breed-screened for FCM (BS-FCM) using an online questionnaire sent to owners of pedigree cats BS-FCM in Sweden, Norway, or Denmark.

## 2. Materials and Methods

### 2.1. Study Design

The study was a questionnaire survey conducted from August to October 2017, based on cardiac examinations conducted between 1 January 2004 to 31 December 2015 in Sweden, Norway or Denmark. Cats that had a minimum of one BS-FCM report in the PawPeds health program for FCM (including an owner signature of release permission) were entered into an electronic data storage. Owners of cats that met the inclusion criteria were sent a questionnaire written in Swedish via email (see Appendix A). This inclusion period was chosen because the screening program for FCM was initiated in 2004, and because cats with their first heart screen performed after 2015 would not have, for the purpose of this study, allowed sufficient follow-up time. The email included information about the study and an URL-link to the questionnaire. The owners were expected to answer the questionnaire within 3 months. One reminder was sent to recipients that had not responded 2 two weeks before the end of the stipulated period.

The study had a second way of inclusion as it was also posted on social media, where owners of cats that had undergone BS-FCM within the PawPeds program were encouraged to answer. The post was written in Swedish and the link to the questionnaire was posted on Facebook at two separate occasions. The first post was shared in 11 Facebook breeding-groups, and the second post was shared 19 times by private accounts and Facebook breeding-groups. The groups were purebred cat-groups in both Sweden and Norway and included “Norsk rasekattforum”, “Sibirisk katt”, “Maine Coon-katten”, “Birmaringen”, “SVERAK-medlemmar på Facebook”, “Birma i fokus”, “Maine Coon Sweden”, “Maine Coon på naturligt vis”, “Perser och Exotic”, “Bengalkatten” and “Birmasällskapet”. The post was also shared to closed groups and accounts which were not presented to us.

### 2.2. The Screening Examination

Cats that undergo screening are pedigree cats without clinical signs of cardiac disease, described according to international guidelines [10]. The BS-FCM examination consists of cardiac auscultation and an echocardiographic examination of a veterinarian linked to the PawPeds program. Today there are 12 veterinarians linked to the program in Denmark, 8 in Norway, and in Sweden there are 22 veterinarians. The screening of cats is according to what is described in the feline consensus for cats [10] and the form for examination is accessible at the PawPeds website (www.pawpeds.com accessed on 10 January 2022). After the echocardiographic examination, the cat is given a classification of normal, equivocal (for left ventricular hypertrophy), HCM (mild, moderate, or severe), RCM or Other (which is further described by the veterinarian performing the examination). All rapports go through a plausibility check by PawPeds before publication in their health register. Current screening recommendation from PawPeds (www.pawpeds.com accessed on 10 January 2022) is to start testing the cats at an age of 1 year and before breeding. Yearly tests are recommended until the cat is more than 3 years old, and thereafter at an age of five. Later testing, for example at the age of 8, should also be considered amongst important breeding cats, cats with close relatives with FCM and amongst cats with equivocal results.

The cardiac reference ranges for the different screening classification are according to previously described recommendations. In brief, it is recommended that a cat could be classified as normal when subjective assessment of cardiac morphology was judged as within the normal variation, which should be supported by a diastolic left ventricular thickness of <5.0 mm in a normally sized cat, i.e., body weight between 2.5–6 kg (if the body weight is outside this range the classification is evaluated on a case-by-case basis) [7,9]. An equivocal classification is considered when the subjective assessment of the echocardiogram does not distinguish a normal phenotype from the mild forms of HCM, when the left ventricular wall thickness is between 5.0 and 5.5 mm in a normally sized cat, i.e., with a body weight of 2.5–6 kg, or both [35], or when the clinical correlate of a specific finding is unknown. A diagnosis of HCM is considered when the subjective assessment of hypertrophy (regional, global or papillary muscle hypertrophy) is supported by a M-mode or 2-dimensional diastolic left ventricular wall thickness of the interventricular septum, left ventricular free wall, or both are measured ≥5.5 mm in a normally sized cat, i.e., with a body weight between 2.5–6 kg [32,36]. A diagnosis of RCM is considered in cats with normal left ventricular wall thickness and atrial enlargement of both or left atrium [37,38]. Restrictive cardiomyopathy was before 2007 classified in the PawPeds program as “Other”. The classification “Other” is specified further by the veterinarian performing the examination, and includes for example congenital heart disease.

### 2.3. The Questionnaire

The questionnaire was designed using the Netigate program (www.Netigate.se accessed on 30 November 2017), and the owners were instructed to respond electronically. The questionnaire was in Swedish and divided into four sections: Section A included general questions; registered name*, identification number, birth date*, breed*, sex, neutering status*, breeding history*, family history of FCM*, and the owners contact information. Parents, littermates and offspring were defined as close relatives in the current study. All owners were instructed to answer section A. Section B concerned cats deceased or lost to follow-up, and section C concerned living cats. Both section B and C included questions about body weight, clinical signs, diagnosis and treatment of heart disease*, response to treatment*, development of CHF and/or ATE*, and the presence of other diseases and diagnostic procedures. Section B further contained questions about the date and cause of death, and the results of post mortem examination*, if performed. Section D was only for cats that had a new owner, or had been relocated to a foster home after the BS-FCM screening, and included information about the new owner so we could find the cat. All information was considered confidential. Mandatory answers in the questionnaire are indicated above with an asterisk (*), and when not answered, the questionnaire was considered incomplete. Screen result outside the inclusion period, incomplete questionnaires, and/or ambiguous results between different screenings led to exclusion of the cat from the study. Screen result outside the inclusion period was seen for owners who read the post on social media and answered the questionnaire online although the cat did not have a screen result during 1 January 2004 and 31 December 2015. Additional screen results after 2015 for cats which met the inclusion criteria were also included. Ambiguous results were defined as having several different conflicting classifications at different screens.

### 2.4. Data Management

All of the data received was entered into an electronic data storage (Microsoft Excel), and then exported into statistical software (JMP v13.0, SAS, Cary, NC, USA) for further analyzes. The cat’s identification number gave the opportunity to go back into the list exported from the PawPeds health register and retrieve the cat’s screening results. The included cats were divided into four groups based on the results from the breed-screens (BS-FCM classification group), according to previously validated classifications: normal (Screen^N^), equivocal (Screen^EQ^), Other (Screen^Other^) or HCM/RCM (Screen^FCM^). Cats that had undergone several BS-FCM screenings with different results were classified according to the classification of the last breed-screening (available at the time of the study, i.e., October 2017).

The cause of death in the survival analyzes was classified as cardiac related or non-cardiac related. Death or euthanasia caused by FCM (including CHF or ATE) was classified as a cardiac related death. All other causes of death or euthanasia were classified as non-cardiac related. When the date of death was reported by the owner as a year or a month, the date of death was set in the middle of the reported period. Deceased cats for which the owner had not reported the date of death were censored from survival analyzes.

### 2.5. Statistical Analyzes

The data was analyzed using commercially available statistical software (JMP v13.0, SAS, Cary, NC, USA) The data is presented as descriptive statistics. Differences between the groups in proportions and for continuous data were analyzed using the Chi-square and Kruskal–Wallis tests, respectively. In the survival analyzes, a log rank test with right censoring was used to determine whether a significant difference existed between the cats that were diagnosed with FCM in the screening program or with an owner reported FCM diagnosis at a later time-point (cats that developed FCM at any time-point) versus those that did not develop FCM at any time-point. Kaplan–Meier curves were used to estimate the median survival time for each diagnostic group and to plot “time to event” curves. The level of significance was set at *p* < 0.05.

## 3. Results

### 3.1. The Study Population

A total of 13,114 cats underwent BS-FCM within the PawPeds health program in Sweden, 11,021/13,114 (84.0%), Norway, 379/13,114 (2.9%) or Denmark, 1714/13,114 (13.1%) during the study period. Contact information was available for the owners of 11,123/13,114 (84.8%) of these cats, and an email with a link to the online questionnaire was sent to these owners. A total of 1200/11,123 (10.8%) questionnaire responses were obtained (Figure 1). In addition, 142 questionnaires were obtained through social media. A total of 229/1342 (17.1%) cats were excluded from the study because they did not meet the inclusion criteria (no screening result within the stipulated time (96/229, 41.9%), incomplete questionnaire (113/229, 49.3%), or because of ambiguous results between screenings (20/229, 8.7%)). The final study population consisted of 1113 cats (Figure 1). Summary statistics of included cats are presented in Table 1 and Table 2, and of these cats, 1029/1113 (92.5%) had been screened in Sweden, 46/1113 (4.1%) in Denmark and 38/1113 (3.4%) in Norway.

### 3.2. PawPeds Screening Results

Based on the PawPeds records, 1011/1113 (90.8%) of the cats in the study population had been classified as Screen^N^, 59/1113 (5.3%) cats as Screen^FCM^, 32/1113 (2.9%) cats as Screen^EQ^ and 11/1113 (1.0%) cats as Screen^Other^ (Figure 1). Of the 59 cats classified as Screen^FCM^ 3/59 (5.1%) cats had RCM-phenotype and 56/59 (94.9%) HCM-phenotype, whereas none were reported to present with the dilated or arrhythmogenic right ventricular cardiomyopathy phenotypes. One cat was classified as Screen^Other^ before 2007 because of congenital heart disease, which, accordingly, did not impact the results. The number of BS-FCM screenings performed per cat in the total study population is presented in Table 3. The median age at the last screening for Screen^N^ was 3.0 (IQR 1.7–4.8) years, Screen^FCM^ 4.3 (IQR 2.4–6.4) years, Screen^EQ^ 3.4 (IQR 2.1–5.1) years and Screen^Other^ 5.1 (IQR 2.6–8.3) years.

### 3.3. Cardiac Related Morbidity

In the total study population, 108/1113 (9.7%) cats were reported to have developed cardiac disease (Figure 2) at some time-point, of which 104 had developed FCM. In the remaining 4 cats (1 Screen^Other^ and 3 Screen^N^), the type of cardiac disease was unspecified. Feline cardiomyopathy was reported in the questionnaire responses for cats of all BS-FCM classification groups. For 9 of 11 cats with a Screen^Other^ result from BS-FCM, the owner reported that the cat did not have a cardiac disease.

Of the 104/1113 (9.3%) cats that had been diagnosed with FCM at breed-screens or at a later time-point, 86/104 (82.7%) had undergone 1 or 2 BS-FCM screenings, and the rest had 3 or more (Figure 3). Of the 45/104 (43.3%) cats diagnosed outside the screening program, 35/45 (77.8%) had been BS-FCM as Screen^N^, 9/45 (20.0%) Screen^EQ^ and 1/45 (2.2%) as Screen^Other^, but had, according to their owners, developed FCM at a time-point after the last BS-FCM. The owners reported (in the questionnaire) the FCM phenotype for the 45 cats that had developed FCM after the last BS-FCM as follows: 21/45 (46.7%) cats with HCM, 6/45 (13.3%) cats with RCM, and 18/45 (40.0%) cats were not defined further than HCM/RCM. Including these numbers with the screen reports from the BS-FCM, HCM was seen in 79/104 (76.0%) of the cats, RCM in 7/104 (6.7%) of the cats, and HCM/RCM in 18/104 (17.3%) of the cats. A higher proportion of males developed FCM compared with the females (*p* < 0.0001). Summary statistics for the cats which were reported to have developed FCM at any time-point are presented in Table 4. A higher proportion of Screen^EQ^ cats were reported to have developed FCM at a later occasion compared to those with a Screen^N^ or Screen^Other^ result (*p* < 0.001). Table 5 shows the number of cats within each BS-FCM classification group that developed FCM at any time-point. 

The age at diagnosis of FCM at any time-point was provided for 79/104 cats (76.0%) that developed FCM, and the median age at diagnosis was calculated for these cats (Table 5). Cats with Screen^N^ had a higher age at the time of diagnosis compared to cats Screen^FCM^ (*p* = 0.03). A total of 7/79 cats (8.9%) were younger than 1 year and 6/79 cats (7.6%) were older than 8 years when diagnosed with a FCM phenotype. The youngest cat was 3.7 months (0.3 years) and the oldest was 180 months (15 years). 

In total, 47/104 (45.2%) of the cats that were diagnosed with or later developed FCM according to the questionnaire responses were reported to have close relatives with FCM. Of the remaining cats that were never reported with a FCM diagnosis, 123/1009 cats (12.2%) were reported to have close relatives diagnosed with FCM, a proportion smaller than corresponding proportion for the cats that developed FCM (*p* < 0.0001). 

### 3.4. Non-Cardiac Related Morbidity

The five most commonly owner-reported non-cardiac diseases (Figure 4) in the study were dental disease, chronic joint disease, neoplastic disease, kidney disease, and infectious/inflammatory conditions. Of the 139/1113 cats with reported non-cardiac disease, 23/139 (16.5%) were reported to also have developed FCM at any time-point. The proportion of non-cardiac morbidities was similar in cats that developed FCM at any time-point compared to cats without an FCM diagnosis.

### 3.5. Survival

Of the cats included in the study, follow-up information concerning date of death was missing in 37/1113 (3.3%) cats, leaving the study population for survival analysis at 1076/1113 (96.7%) cats. The median follow-up time after breed-screen was 3.3 years (IQR 1.39–6.4) with a median age at follow-up of 7.4 years (IQR 4.8–10.4). Out of the population analyzed, 312/1076 (29.0%) cats were reported dead at a median age of 8.3 years (IQR 4.7–13.4). Kaplan–Meier survival curves for subgroups based on BS-FCM classification are shown in Figure 5. Cats classified as Screen^FCM^ had a shorter life span compared to cats classified as Screen^N^ (Figure 5a, *p* < 0.001). For cardiac related death, cats classified as Screen^FCM^ had shorter survival times than those with other screen results (Figure 5b, *p* < 0.0001). Cats that developed FCM at any time-point had a shorter life span compared to cats reported not to have developed FCM (median 9.2 years, IQR 4.8–14.4 vs. 13.4 years, IQR 10.6–16.2, *p* < 0.0001, Figure 5c).

### 3.6. Overall Mortality

The most common causes of death in the total study population were kidney/urinary tract disease (55/312, 17.6%), cardiac disease (51/312, 16.3%), neoplastic disease (47/312, 15.1%), and infectious disease (35/312, 11.2%) (Figure 6).

### 3.7. Cardiac Related Morbidity and Mortality

A cardiac related death was reported in 45/58 (77.6%) of the deceased cats that developed FCM at any time-point, a proportion higher than the proportion for cats reported not to have developed FCM (*p* < 0.0001). Of the cats with a cardiac related death after developing FCM at any time-point, 12/45 (26.7%) were reported to have developed CHF, and 11/45 (24.4%) were reported to have developed ATE as a consequence of FCM. The remaining 22/45 (48.9%) cats were reported dead due to FCM but the cause of death was not further specified. In the total study population, 16/1113 cats (1.4%) were reported to have developed ATE of which 13/16 (81.3%) were reported to have a previous diagnosis of FCM. Three cats in the study were reported to have developed both CHF and ATE and all of these cats had been classified as Screen^FCM^ in PawPeds records. A higher proportion of cats that developed FCM developed ATE compared to the proportion of those without FCM (*p* < 0.0001).

## 4. Discussion

This is the first published questionnaire based long-term follow-up study concerning BS-FCM cats. A diagnosis of FCM were given at the first breed-screen for approximately 50% of the cats with a Screen^FCM^-result, and more cats would likely have been identified within the screening program if repeated breed-screens at a higher age had been performed in more cats. Cats that developed FCM at any time-point had a shorter lifespan and survival time than those without FCM. A higher risk to die a cardiac related death was seen for cats that developed FCM at any time-point, compared to those that did not. Non-cardiac morbidities were similar in nature and proportion for cats with and without FCM.

The incidence of FCM, developed at any time-point, based on the results from owners’ questionnaires, was 9.3% in the present study population, which is a low proportion in comparison to previous studies; these have a reported prevalence of 14.7 and 15.5%, respectively [5,6]. However, the present study included a larger study population in comparison to studies performed previously, and only included pedigree cats. Because the majority of cats (44.5%) in the present study were only BS-FCM at one occasion, and often early in life (as most cats are screened at 1–2 years of age according to the screening recommendation by PawPeds), it is possible that some cats had very mild preclinical FCM that could not be detected by echocardiography and these cats developed overt FCM later in life. Furthermore, 25.0% of the cats in the study were less than 4.8 years old at follow-up, so the proportion of cats developing FCM over time might be higher than the present result. The most common owner-reported FCM phenotype in the present study was HCM, which is in agreement with previous published studies [5,6,7,8,16,17,21,22,23,24,25,26,27,28,29,30,31].

Our study suggests that repeated breed-screens at a higher age result in an increased likelihood of detecting FCM, as some cats, despite previous unremarkable BS-FCM results, developed FCM later in life according to the owners. Feline cardiomyopathies are acquired diseases that can develop and become clinically detectable later in life. The median age at diagnosis for cats that developed FCM at any time-point was 3.73 years. This is lower than previous reports, which have reported the median age at diagnosis for HCM to be 5–7 years [21,24,28], and a possible explanation is that these cats were younger because they were apparently healthy cats screened before breeding. This is a clear indication that repeated breed-screenings at a higher age should be recommended, which reinforces the current breeding recommendations by PawPeds [32]. In total, 8.9% of the cats were younger than 1 year when they developed FCM. Based on these results, it appears appropriate to commence screening from an age of 1 year, or prior to breeding the cats, which is in accordance with the current breed-screening recommendation [32]. A total of 7.6% of the cats developed FCM at an age higher than 8 years. This suggests that particularly important breeding cats, and cats within breeding lines where FCM exists, should preferably undergo screening at an age older than 8 years. It is important to identify individuals affected with FCM, not just for breeding concerning that specific individual, but for breeding of that individual’s close relatives, due to HCM’s hereditary nature. A significantly higher proportion of cats with a Screen^EQ^-result developed FCM, compared to those with a screening diagnosis of Screen^N^ or Screen^Other^. This implies that cats with an equivocal result should be re-examined before breeding which is in agreement with the current recommendations.

Cats that developed FCM were significantly more likely to have reported close relatives with the same diagnosis, compared to cats that did not develop FCM. Previous studies have described HCM as a familial disease inherited as an autosomal dominant trait [29,33,34], and the results of this study, although not designed to study the mode of inheritance, are not at odds with a genetic background for FCM. Individuals with close relatives with a diagnosis of FCM are, therefore, of particular interest for repeated breed-screens at a higher age. A significantly higher proportion of males developed FCM compared with females. This is in accordance with previous studies [6,8,13,14,27,31].

Non-cardiac morbidities were similar in nature and in proportion in cats that developed FCM at any time-point compared to cats without a FCM diagnosis. The most commonly reported non-cardiac diseases were dental disease, chronic joint disease, neoplastic disease, kidney disease, and infectious/inflammatory conditions. This result differs from a previous morbidity study, where the three most common morbidities were trauma and gastrointestinal and lower urinary tract problems [2]. This morbidity study was based on actuarial data, and many common co-morbidities lead to costs that are partly covered or not covered at all by an insurance policy, which means that they are not fully included in the actuarial data as these costs may not exceed the insurance deductible. Such conditions include some types of dental procedures and chronic joint disease, which is often responsive to analgesic therapy and might not always be fully investigated.

In the current study, cats that had developed FCM at any time-point had a significantly higher risk of developing ATE than cats without FCM. Previous studies have also reported a clear association between feline HCM and the development of ATE [14,18,19,26]. A significant association between an owner stated development of FCM and a cardiac-related death was found, although, only 12/45 (26.7%) of these cats were reported to have developed CHF and 11/45 (24.4%) to have developed ATE. Accordingly, there was a discrepancy between cats developing common clinical signs of heart disease, such as CHF and ATE, and the proportion of owner-reported cardiac related deaths. Many BS-FCM cats belong to a breeding population and potentially some of these cats may have been euthanized because they could no longer be used for breeding. The proportion of cats with FCM that did develop CHF and ATE was similar to what was seen in a previous study [14] which reported 24.2% and 11.6%, respectively. In that study the cardiac related deaths were 27.9% for cats with hypertrophic cardiomyopathy, a number considerably smaller than this study, which reported a cardiac related death in 45/58 (77.6%) of the cats that developed FCM at any time-point. Furthermore, cats with FCM had a significantly shorter lifespan than those without. This was also found in a previous study [4], where a significantly shorter survival time was seen for all-cause survival in cats with preclinical HCM due to cardiovascular death compared to apparently healthy cats. 

The most common causes of death in the overall mortality were kidney/urinary, cardiac, neoplastic, and infectious disease. The result is comparable to that from a previous mortality study [1], which reported urinary/kidney, traumatic, neoplastic, and infectious and cardiac disease as the most common causes of death. The median survival time for cats that developed FCM at any time-point was 9.2 years (IQR 4.8–14.4). This means that the cats in the present study that developed FCM at any time-point were comparably young at death, compared to what was found in the above-mentioned mortality-study, where a median survival time for life-insured cats was reported to be >12 years [1]. Again, it cannot be excluded that some cats in the present study had been euthanized due to lack of breeding potential following a FCM diagnosis.

The study had some limitations: The study was dependent on the owner’s willingness to participate and their recollection of events. The benefit of a questionnaire study is the ability to reach a large number of cat owners, but the drawback is the limited resolution in data. The present study had a response rate of 10.8%. Because the study involved a rather large study population and spanned over 12 years, we consider the overall response rate acceptable. However, the follow-up time for individuals was restricted to a median follow-up time of 3.3 years (IQR 1.39–6.4) after breed-screen. Ideally, all cats should have been followed until they died. The majority of cats included in this long-term outcome questionnaire study were BS-FCM in Sweden, which can be explained by the fact that 84.0% of the cats were BS-FCM in Sweden. The questionnaire was distributed in Swedish, which could be more difficult to understand for cat owners in the other countries. A questionnaire study is usually vulnerable to subjectivity. Feline cardiomyopathy is a serious, possibly fatal disease in cats, and owners who have experienced FCM in their pets could have more interest to participate than those with cats without FCM. This could result in a larger response rate for cats with cardiac disease, compared to healthy cats. On the other hand, owners could also choose not to answer the questionnaire because the disease is a sensitive matter for many breeders. Many breeders have many cats, and because the questionnaire should be completed for each cat separately, some breeders could have chosen not to respond for time reasons or only responded for affected cats as a priority of information to share. Many of the cats were pedigree cats and the results do not necessarily reflect the general cat population. Furthermore, the number of cats of certain breeds were too few to allow meaningful comparisons of FCM occurrence by breed. A further limitation was that 45/104 (43.3%) of the cats that developed FCM at any time-point had the diagnosis reported by the owners and the diagnosis could not be further verified when reviewing the questionnaires. Owners might have interpreted BS-FCM cats as Screen^EQ^ as cats with FCM, which could have had a slight influence on the study’s owner reported FCM prevalence. The owners for 9 of 11 cats with a Screen^Other^ result from BS-FCM reported that the cat did not have a cardiac disease, which might also have had a small influence on the study result. A further limitation was that the study did not exclude secondary FCM in the elderly cats, which limits the statement that cats older than 8 years should undergo further screenings.

## 5. Conclusions

Approximately 50% of the cats diagnosed with FCM within the screening program had been diagnosed at the first BS-FCM. The large proportion of cats that developed FCM at a later time-point underscores the need for repeated breed-screenings later in life. Cats that developed FCM at any time-point had a shorter life span, with a similar proportion and in line with the nature of non-cardiac morbidities, compared to those without FCM.

## Figures and Tables

**Figure 1 animals-12-02782-f001:**
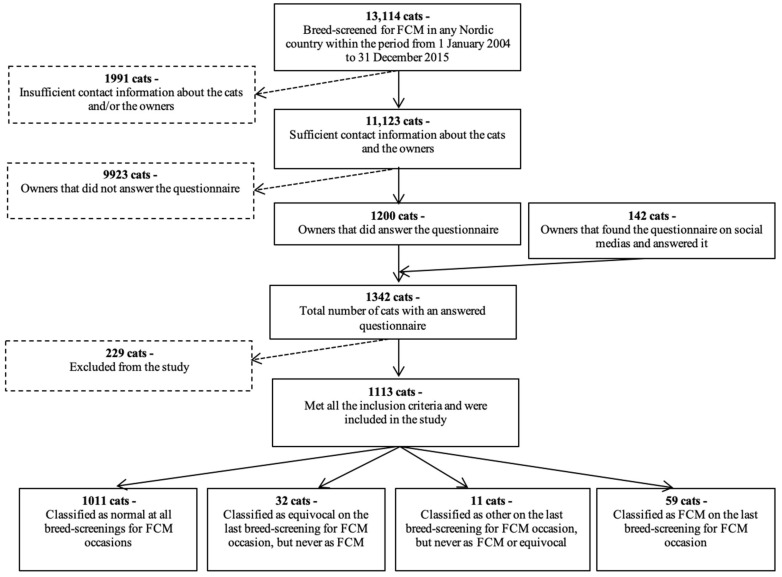
Flow chart describing reasons for excluding cats to create the study population.

**Figure 2 animals-12-02782-f002:**
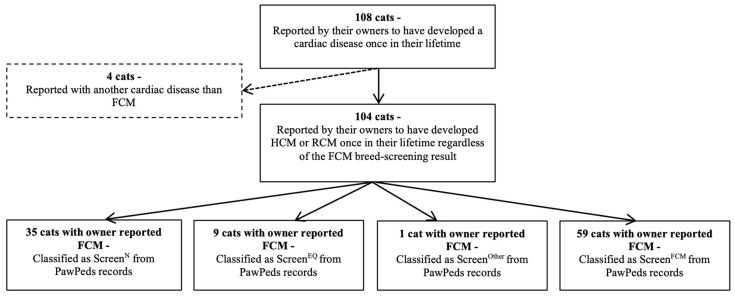
Flow chart describing development of cardiac disease at any time-point in the study population. Screen^N^ = Cats classified as normal at breed-screening, Screen^EQ^ = Cats classified as equivocal at breed-screening, Screen^FCM^ = Cats diagnosed with HCM or RCM at breed-screening, Screen^Other^ = Cats classified as Other at breed-screening.

**Figure 3 animals-12-02782-f003:**
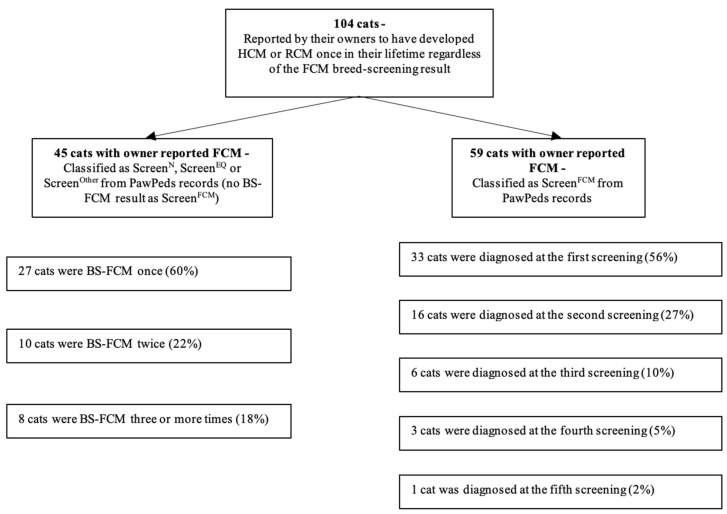
Illustration of numbers of BS-FCM performed per cat that developed FCM at any time-point, regardless of the previous screen result. BS-FCM = Breed-screened for FCM. FCM = feline cardiomyopathy.

**Figure 4 animals-12-02782-f004:**
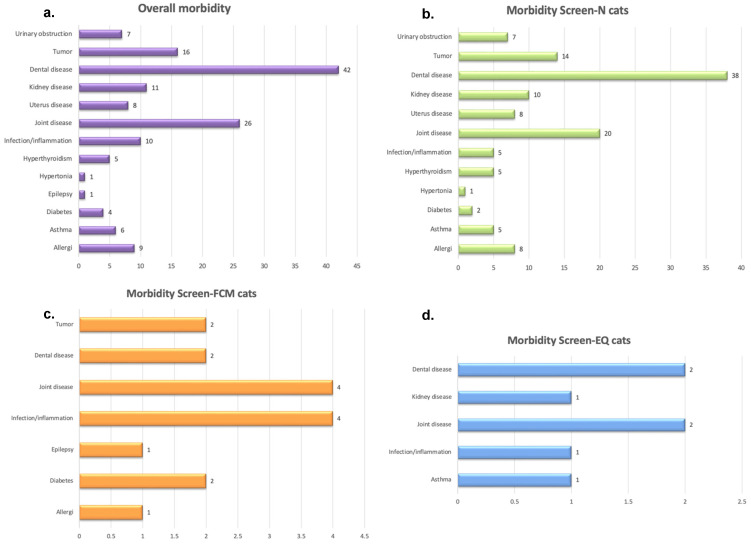
(**a**) Overall non-cardiac morbidity in the total study population reported by owners (n = 139/1113, 12.5%), (**b**) Morbidity in cats with Screen^N^ (n = 118/139, 84.9%), (**c**) Morbidity in cats with Screen^FCM^ (n = 15/139, 10.8%), (**d**) Morbidity in cats with Screen^EQ^ (n = 6/139, 4.3%). No diagram is presented for cats with Screen^Other^, because no extracardiac disease was reported in this group. More than one disease could be reported for each cat. Screen^N^ = Cats classified as normal at breed-screening, Screen^EQ^ = Cats classified as equivocal at breed-screening, Screen^FCM^ = Cats diagnosed with HCM or RCM at breed-screening, Screen^Other^ = Cats classified as Other at breed-screening.

**Figure 5 animals-12-02782-f005:**
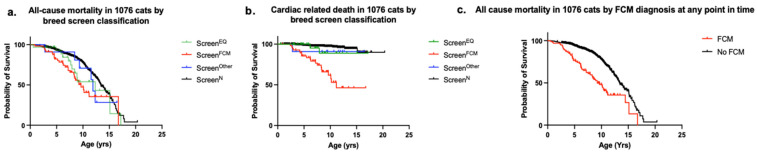
(**a**) All-cause mortality Kaplan–Meier survival curves in 1076 cats by breed-screen classification plotting the estimated percentage of cats that have not yet died against age in cats based on breed-screening results. Cats with Screen^FCM^ had significantly (*p* < 0.0001) shorter life span (n = 57, median 9.6 years (IQR 6.3–16.7)) compared to cats with Screen^N^ (n = 977, median 13.4 years (IQR 10.4–16.1)). Cats with Screen^EQ^ (n = 31) had a median life span of 12.3 years (IQR 7.7–15.2) and cats with Screen^Other^ (n = 11) had a median life span of 11.9 years (IQR 9.3–NA). Right censored data was included and 37 cats out of the 1113 cats were excluded because no date of death was reported. (**b**) Cardiac related death Kaplan–Meier survival curves in 1076 cats by breed-screen classification plotting the estimated percentage of cats that have not yet died against age in cats based on breed-screening results. Cats with Screen^FCM^ (n = 57) had a significantly (*p* < 0.0001) shorter life span compared with cats Screen^N^ (n = 977), Screen^EQ^ (n = 31), or Screen^Other^ (n = 11). The median survival time could only be calculated for cats breed-screened as FCM (11.2 years, IQR 7.5–NA). Right censored data was included and 37 cats out of the 1113 cats were excluded because no date of death was reported. (**c**) All-cause mortality Kaplan–Meier survival curves in 1076 cats by FCM development at any time-point plotting the estimated percentage of cats that have not yet died against age, comparing survival in cats that developed FCM at any time-point (n = 100) versus those that did not (n = 976). Cats that developed FCM at any time-point had a significantly (*p* < 0.0001) shorter life span (median 9.2 years, IQR 4.8–14.4) compared to those that did not (median 13.4 years, IQR 10.6–16.2). Right censored data was included and 37 cats out of the 1113 cats were excluded because no date of death was reported. FCM = feline cardiomyopathy. Screen^N^ = Cats classified as normal at breed-screening, Screen^EQ^ = Cats classified as equivocal at breed-screening, Screen^FCM^ = Cats diagnosed with HCM or RCM at breed-screening, Screen^Other^ = Cats classified as Other at breed-screening.

**Figure 6 animals-12-02782-f006:**
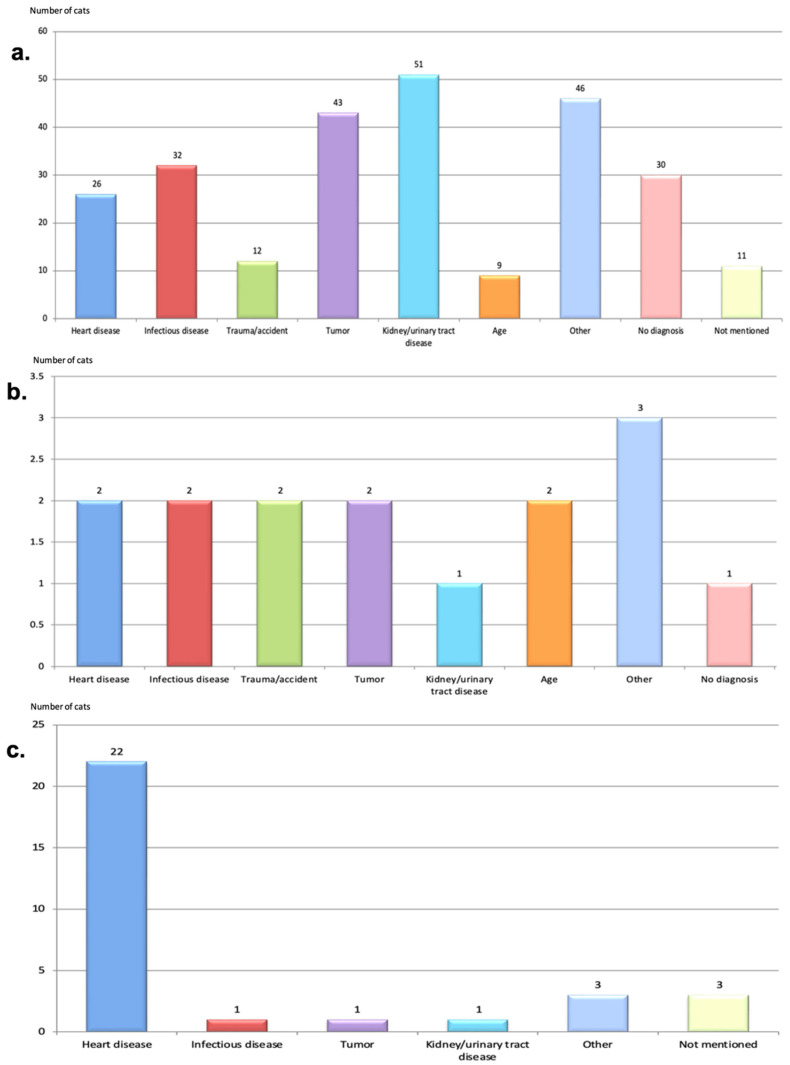
(**a**) The proportional mortality in deceased Screen^N^ cats (n = 260), (**b**) The proportional mortality in deceased Screen^EQ^ cats (n = 15), (**c**) The proportional mortality in deceased Screen^FCM^ cats (n = 31), (**d**) The proportional mortality in deceased Screen^Other^ cats (n = 6), (**e**) The proportional mortality for all deceased breed-screened cats (n = 312). Examples of “other” diseases includes chronic eye disease, spondylosis, allergy, epileptic seizures, surgical disease. Screen^N^ = Cats classified as normal at breed-screening, Screen^EQ^ = Cats classified as equivocal at breed-screening, Screen^FCM^ = Cats diagnosed with HCM or RCM at breed-screening, Screen^Other^ = Cats classified as Other at breed-screening.

**Table 1 animals-12-02782-t001:** Summary statistics for the 1113 cats included in the study by sex, breed-screen classification, and breed.

Breed	Male (n = 408)				Female (n = 705)				Total
	Screen^EQ^ n = 16	Screen^FCM^ n = 42	Screen^N^ n = 348	Screen^Other^ n = 2	Screen^EQ^ n = 16	Screen^FCM^ n = 17	Screen^N^ n = 663	Screen^Other^ n = 9	
Maine Coon	7/223 (3.1%)	5/223 (2.2%)	75/223 (33.6%)		3/223 (1.3%)	1/223 (0.4%)	132/223 (59.2%)		223
Siberian		8/205 (3.9%)	65/205 (31.7%)	1/205 (0.5%)	2/205 (1.0%)	1/205 (0.5%)	125/205 (61.0%)	3/205 (1.5%)	205
Norwegian Forest	3/201 (1.5%)	5/201 (2.5%)	72/201 (35.8%)		2/201 (1.0%)	2/201 (1.0%)	116/201 (57.7%)	1/201 (0.5%)	201
Birman		3/126 (2.4%)	37/126 (29.4%)		1/126 (0.8%)	1/126 (0.8%)	80/126 (63.5%)	4/126 (3.2%)	126
British shorthair	3/87 (3.4%)	6/87 (6.9%)	19/87 (21.8%)		1/87 (1.1%)		57/87 (65.5%)	1/87 (1.1%)	87
Cornish Rex		5/58 (8.6%)	14/58 (24.1%)		2/58 (3.4%)	5/58 (8.6%)	32/58 (55.2%)		58
Ragdoll		1/53 (1.9%)	12/53 (22.6%)			1/53 (1.9%)	39/53 (73.6%)		53
Sphynx	1/42 (2.4%)	3/42 (7.1%)	17/42 (40.5%)	1/42 (2.4%)	1/42 (2.4%)	4/42 (9.5%)	15/42 (35.7%)		42
Devon Rex	2/37 (5.4%)	2/37 (5.4%)	10/37 (27.0%)			1/37 (2.7%)	22/37 (59.5%)		37
Bengal		1/30 (3.3%)	13/30 (43.3%)				16/30 (53.3%)		30
Other breed		3/51 (5.9%)	14/51 (27.5%)		4/51 (7.8%)	1/51 (2.0%)	29/51 (56.9%)		51
Total	16/1113 (1.4%)	42/1113 (3.8%)	348/1113 (31.3%)	2/1113 (0.2%)	16/1113 (1.4%)	17/1113 (1.5%)	663/1113 (59.6%)	9/1113 (0.8%)	1113

Percentage reflects the proportion of cats within each breed. Breeds with less than thirty cats included were collapsed into one category “other breed” which included cats of 15 breeds; Abyssinian, Burmese, Don Sphynx, European, Exotic, Korat, LaPerm, Manx, Neva masquerade, Oriental shorthair, Persian, Russian Blue, Siamese, Turkish Van, Non-recognized shorthair. Screen^N^, Cats classified as normal at breed-screen for FCM; Screen^FCM^, Cats diagnosed with HCM or RCM at breed-screen for FCM; Screen^EQ^, Cats classified as equivocal at breed-screen for FCM; Screen^Other^, Cats classified as Other at breed-screen for FCM; FCM, feline cardiomyopathy.

**Table 2 animals-12-02782-t002:** Distribution of the number of cats alive, dead, or lost to follow-up at the time of the questionnaire by breed-screen classification for the total study population.

Classification	Dead (n = 312)	Lost to Follow-Up (n = 4)	Alive (n = 797)	Total (n = 1113)
Screen^N^	260/1011 (25.7%)	3/1011 (0.3%)	748/1011 (74.0%)	1011
Screen^FCM^	31/59 (52.5%)		28/59 (47.5%)	59
Screen^EQ^	15/32 (46.9%)		17/32 (53.1%)	32
Screen^Other^	6/11 (54.5%)	1/11 (9.1%)	4/11 (36.4%)	11
Total	312/1113 (28.0%)	4/1113 (0.4%)	797/1113 (71.6%)	1113

Percentages reflects the proportion of cats within each breed-screen classification. Screen^N^, Cats classified as normal at breed-screen for FCM; Screen^FCM^, Cats diagnosed with HCM or RCM at breed-screen for FCM; Screen^EQ^, Cats classified as equivocal at breed-screen for FCM; Screen^Other^, Cats classified as Other at breed-screen for FCM; FCM, feline cardiomyopathy.

**Table 3 animals-12-02782-t003:** Number of breed-screens performed per cat in the total study population, median age and interquartile range, and number and proportion of cats with a Screen^FCM^ result.

Number of BS-FCM	Number and Proportion of Cats	Median Age (IQR)	Number and Proportion of Cats with a Screen^fcm^ Result
1	495/1113 (44.5%)	1.6 (1.1–2.8)	33/59 (55.9%)
2	329/1113 (29.6%)	3.2 (2.5–4.5)	16/59 (27.1%)
3	165/1113 (14.8%)	4.4 (3.4–5.5)	6/59 (10.2%)
4	81/1113 (7.3%)	5.8 (5.1–7.1)	3/59 (5.1%)
5	35/1113 (3.1%)	7.7 (6.4–8.4)	1/59 (1.7%)
6	6/1113 (0.5%)	7.3 (6.4–10.0)	-
7	2/1113 (0.2%)	4.8 (3.6–5.9)	-
Total	1113/1113 (100%)		59/59 (100%)

BS-FCM, Breed-screen for feline cardiomyopathy; IQR, interquartile range; Screen^FCM^; Cats diagnosed with HCM or RCM at breed-screen for FCM. Note: The table is not cumulative, which means that a cat is only counted once and only occurs in one row.

**Table 4 animals-12-02782-t004:** Summary statistics of sex for the 104 cats that developed feline cardiomyopathy at any time-point by breed, with regards to the 1113 cats included in the study.

Breed	Male (n = 64)	Female (n = 40)	Total (n = 104)
Maine Coon (n = 223)	9	9	18
Siberian (n = 205)	10	4	14
Norwegian Forest (n = 201)	7	3	10
Birman (n = 126)	7	5	12
British Shorthair (n = 87)	12	1	13
Cornish Rex (n = 58)	6	8	14
Ragdoll (n = 53)	1	2	3
Sphynx (n = 42)	4	4	8
Devon Rex (n = 37)	3	1	4
Bengal (n = 30)	1	1	2
Persian (n = 17)	1	2	3
Exotic (n = 8)	1	0	1
European (n = 5)	1	0	1
Siamese (n = 1)	1	0	1
Total	64/408 (15.7 %)	40/705 (5.7 %)	104/104 (100%)

**Table 5 animals-12-02782-t005:** Proportion of the 104 cats that was reported to have developed feline cardiomyopathy at any time-point, the owner-reported median age and interquartile range at diagnosis of FCM by breed-screen classification, and age distribution for the last breed-screen.

Classification	Proportion of Risk Population	Median Age (Years)	IQR (Years)	Age Distribution of the Last BS-FCM (Median and IQR) (Years)
Screen^N^ (n = 1011)	35/1011 (3.5%)	4.65 (n = 16)	3.77–7.53	2.5 (1.4–3.1)
Screen^FCM^ (n = 59)	-	3.40 (n = 59)	2.10–5.10	4.3 (2.4–6.4)
Screen^EQ^ (n = 32)	9/32 (28.1%)	4.01 (n = 4)	1.21–5.65	3.4 (4.0–4.9)
Screen^Other^ (n = 11)	1/11 (9.1%)	- (n = 0)	-	2.5 (NA)
Total	104/1113 (9.3%)	3.73 (n = 79)	2.20–5.35	-

FCM, feline cardiomyopathy; Screen^N^, Cats classified as normal at breed-screen for FCM; Screen^FCM^, cats diagnosed with HCM or RCM at breed-screen for FCM; Screen^EQ^, Cats classified as equivocal at breed-screen for FCM; Screen^Other^, Cats classified as Other at breed-screen for FCM.

## Data Availability

The data presented in this study are available on request from the corresponding author.

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
