# Peer review of "A Questionnaire Survey on Long-Term Outcomes in Cats Breed-Screened for Feline Cardiomyopathy"

_animals, 2022, doi:10.3390/ani12202782_

Round 1

Reviewer 1 Report

Follby et al. submitted a descriptive study to explore the long-term outcomes for cats breed-screened for FCM. Overall, the study is well presented and easy to follow. I particularly appreciated how the authors presented how the data were obtained (including exclusions) and the repartition of the cats based on the different criteria (e.g., see Fig. 1). Honestly, my only concern is that the presentation of the results, either in the text or in the tables (and figures) overlaps with the text. Thus, the results are often the same information as what is already available in the tables or figures. Finally, the conclusion that additional screening is necessary is good advice. However, I feel like this conclusion could have been drawn without the current study's data.

Author Response

Thank you for your work reviewing our article. We have removed abundant text and made reference to appropriate figures. Please see the paragraph starting at line 247.

We agree that the conclusion that repeated screenings might appear obvious. However, we are not aware of any other study that has actually previously shown this. Furthermore, it is a problem to convince cat breeders to repeat screenings up to 7-8 years of age. Therefore, we would very much like to keep this conclusion. 

Reviewer 2 Report

Well written article. Very interesting given the large number of cases. The large time interval of the study allows you to have an overview of one of the most common diseases in cats. The study design is well explained. The results presented clearly and comprehensively. The statistical study is adequate and supports the results. Well-articulated discussions presented in an easy-to-follow way.

Author Response

Thank you for your work reviewing our article. We are uncertain if the reviewer wants us to edit the English language according to the ticked box, and no guideline concerning the language were included for the Authors. Two of the other reviewers considered the English language fine or with minor spell checks, and professional English editing has been performed. Also, this reviewer starts the comments and suggestions for Authors with compliments for a well written article.

Reviewer 3 Report

When you refer to the storage of the data, please do not call Microsoft Excel a database. It is not. You can speak about an electronic data storage or electronic data notebook, but please try to avoid speaking about "databases".

I would suggest also to avoid pie charts. They are pretty catchy, but humans are quite bad "angle discriminators" thereof it is much better the use of histograms if we want to clearly show the difference between groups.

Author Response

Thank you for your work reviewing our article. We have adjusted the text in the manuscript according to the comments received. Please see line 65, 79, 111, 164 and 167.

We have changed the pie charts into histogram, as suggested. Please see Figure 6. 

Reviewer 4 Report

The submitted manuscript is a well-documented study of the long-term outcome of cats screened for feline cardiomyopathy (BS-FCM) in Sweden, Norway, and Denmark. The manuscript is extremely well-written and addresses a health issue with cats.

All sections of the manuscript are presented in the correct format. The abstract is brief, pertinent, and appropriate for the article. The authors provided a comprehensive overview of feline cardiomyopathy in the Introduction section. Mentioned are the FCM types responsible for this condition. The clinical manifestations of this disease are elucidated in order to make the presentation of the study pertinent.

The Materials and Methods section describes in depth the questionnaires used to collect data and the statistical techniques employed.

Both tables and comments provide a comprehensive presentation of the results. The discussion section compares the results of the study to those of other reports, and the conclusions support the presented findings.

The topic of the manuscript, which is a health concern in Nordic countries, makes for an interesting read.

Due to its precision and well-executed presentation, the manuscript meets the requirements for publication in your Journal.

Author Response

We thank this reviewer for kind words and review of our paper. There is a discrepancy between the boxes ticked and the comments and suggestions for Authors. We are uncertain if we are expected to revise, and if so, what, we need to revise.